# Probing polaron-induced exciton quenching in TADF based organic light-emitting diodes

Monirul Hasan [1,2], Siddhartha Saggar [1,2], Atul Shukla [1,2], Fatima Bencheikh[3], Jan Sobus[1,2], Sarah K. M. McGregor [2,4], Chihaya Adachi [3✉], Shih-Chun Lo [2,4✉] & Ebinazar B. Namdas [1,2✉]

Polaron-induced exciton quenching in thermally activated delayed fluorescence (TADF)-based organic light-emitting diodes (OLEDs) can lead to external quantum efficiency (EQE) roll-off and device degradation. In this study, singlet-polaron annihilation (SPA) and triplet-polaron annihilation (TPA) were investigated under steady-state conditions and their relative contributions to EQE roll-off were quantified, using experimentally obtained parameters. It is observed that both TPA and SPA can lead to efficiency roll-off in 2,4,5,6-tetra(9H-carbazol-9-yl)isophthalonitrile (4CzIPN) doped OLEDs. Charge imbalance and singlet-triplet annihilation (STA) were found to be the main contributing factors, whereas the device degradation process is mainly dominated by TPA. It is also shown that the impact of electric field-induced exciton dissociation is negligible under the DC operation regime (electric field < 0.5 MV cm$^{-1}$). Through theoretical simulation, it is demonstrated that improvement to the charge recombination rate may reduce the effect of polaron-induced quenching, and thus significantly decrease the EQE roll-off.

[1] School of Mathematics and Physics, The University of Queensland, Brisbane, QLD 4072, Australia. [2] Centre for Organic Photonics & Electronics, The University of Queensland, Brisbane, QLD 4072, Australia. [3] Center for Organic Photonics and Electronics Research, Kyushu University, Fukuoka 819-0395, Japan. [4] School of Chemistry and Molecular Biosciences, The University of Queensland, Brisbane, QLD 4072, Australia. ✉email: adachi@cstf.kyushu-u.ac.jp; s.lo@uq.edu.au; e.namdas@uq.edu.au

Organic light-emitting diodes (OLEDs) have shown significant improvement in device efficiency within the last decades and are now extensively implemented for display and illumination applications. Commercial small-area non-transparent displays generally need a minimum brightness level of ~500 cd m$^{-2}$, while transparent displays require a brightness level of more than 5000 cd m$^{-2}$ in order to overcome photo-interference from ambient sunlight[1,2]. Similarly, high brightness is also essential for general illumination applications[1,3]. Many OLEDs have shown very high external quantum efficiencies (EQEs ~20–30%) with brightness below 100 cd m$^{-2}$. However, the EQE drops significantly for brightnesses over 1000 cd m$^{-2}$. This EQE roll-off is associated with not only high-power consumption in OLEDs but also device degradation under elevated current densities.

OLEDs with an internal quantum efficiency (IQE) of nearly 100% can be obtained by using phosphorescence and thermally activated delayed fluorescence (TADF)-based emitters. Despite the high IQEs, both phosphorescent[4,5] and TADF[6–8] type emitters exhibit extensive EQE roll-off at high current densities, which is often required for high brightness. This decrease in EQE is predominately attributed to exciton quenching through exciton–exciton and exciton–polaron interactions. It is suggested that the efficiency roll-off in phosphorescent OLEDs is predominately due to the long lifetime of triplets—usually in the order of microseconds to milliseconds. This can lead to excessive accumulation of triplet excitons under steady-state OLED operations and eventually, density-driven loss mechanisms such as triplet–triplet annihilation (TTA) and triplet–polaron annihilation (TPA)[9–16]. In TADF emitters, triplet harvesting can occur through reverse intersystem crossing (RISC) of triplets to singlet states via thermal activation. This RISC process takes place mostly over a microsecond time regime. As a result, TADF OLEDs operated at high current densities can accumulate triplets which may give rise to singlet–triplet annihilation (STA), TTA, and TPA. Several reports have shown the existence of STA and TTA in TADF systems under optical and electrical excitation[17–21]. Moreover, Sandanayaka et al.[22] have shown that not only triplet interaction-based annihilation processes, but also singlet–polaron annihilation (SPA) can significantly reduce singlet population. In addition, TPA and TTA were identified as major contributors to device degradation, carrier trap states generation, and charge imbalance in the TADF OLEDs[23,24]. However, the impact of polaron-induced quenching and electric field-induced exciton dissociation, along with disentangling the role of different loss processes in EQE roll-off for TADF OLEDs have not yet been studied.

In this work, the EQE roll-off for TADF OLEDs utilizing 2,4,5,6-tetra(9H-carbazol-9-yl)isophthalonitrile (4CzIPN) as the emitter and 1,3-bis(N-carbazolyl)benzene (mCP) as the host was investigated (Supplementary Fig. 1). SPA and TPA rates were quantified using steady-state photoluminescence (PL) and electroluminescence (EL) measurements. Field-induced quenching rate was measured independently from singlet exciton binding energy. Further, the influence of the different loss mechanisms was probed by considering their relevant interplay in the TADF process. Subsequently, strategies to reduce the effect of polaron-induced quenching and EQE roll-off have also been discussed briefly.

## Results

To study the effect of polaron-induced quenching in TADF OLEDs, we have fabricated single-carrier hole-only devices (HODs) with 5 wt% 4CzIPN doped in mCP host (device structure in Supplementary Fig. 2). A dual excitation setup was used for this study, involving optical and electrical excitation under steady-state conditions (Supplementary Fig. 3). The optical excitation power was kept at the constant value of 50 µW throughout the experiment to limit the effect of exciton–exciton interaction. This experiment allowed us to record PL intensities under various applied voltages and current densities. The PL intensities under different bias voltages are shown in Supplementary Fig. 4. The PL intensity drop with the increase in voltage can be attributed to the SPA, TPA, and field-induced quenching processes. The time evolution of singlet and triplet population considering these exciton annihilation processes can be modeled for the dual excitation system as

$$\frac{dS}{dt} = I_x - k_S S - k_{ISC} S + k_{RISC} T - k_{SP} SP - R(f)S, \quad (1)$$

$$\frac{dT}{dt} = -k_{RISC} T - k_T T + k_{ISC} S - k_{TP} TP - R(f)T, \quad (2)$$

where $S$, $T$, and $P$ represent the singlet, triplet, and polaron densities, respectively; $I_x$ represent the exciton generation rate from optical excitation; $k_S$, $k_{ISC}$, $k_{RISC}$, and $k_T$ are the singlet decay rate constant from the singlet excited state to the ground state (sum of radiative and non-radiative decay rates), intersystem crossing (ISC) rate constant from the singlet state to the triplet state, RISC rate constant from the triplet state to the singlet state, and non-radiative triplet decay rate constant from the triplet excited state to the ground state, respectively; $k_{SP}$, $k_{TP}$, and $R(f)$ refer to the rate constants of SPA, TPA, and electric field ($f$) induced quenching rate, respectively. The PL decay of the blend (Supplementary Fig. 5) shows a fast prompt component with a lifetime of $\tau_P$ ~ 14 ns, and a slower delayed component of $\tau_D$ ~ 2.9 µs. From the PL quantum yield (QY = 84%), the prompt efficiency ($\Phi_P$) and delayed efficiency ($\Phi_D$) can be estimated as 41.9% and 42.1%, respectively. The rate constants of $k_S$, $k_{ISC}$, $k_{RISC}$, and $k_T$ of mCP:4CzIPN blend were calculated from $\tau_P$, $\tau_D$, $\Phi_P$, and $\Phi_D$ (details in Supplementary Note 1, and the values are summarized in Supplementary Table 1). To solve Eqs. (1) and (2), the average polaron densities under different bias voltages were obtained using a one-dimensional drift-diffusion model[25,26] (details in Supplementary Note 2 and the parameters are listed in Supplementary Table 2). Supplementary Fig. 6 shows the experimental and simulated current density–voltage ($J–V$) response of the HOD devices and corresponding spatial polaron distribution used for the calculation of average polaron densities. A small deviation between experimental and simulated current density was observed below 1 V in Supplementary Fig. 6a, potentially due to the leakage current. Hole mobility of $1 \times 10^{-5}$ cm$^2$ V$^{-1}$ s$^{-1}$ and characteristic field of $1.13 \times 10^6$ V cm$^{-1}$ were obtained from the HOD devices.

To investigate the field-induced exciton dissociation in TADF emitter, we fabricated OLED devices employing mCP:5 wt% 4CzIPN as the emissive layer (Fig. 1 inset). In order to avoid EL emission and polaron-induced quenching, the OLEDs were tested under reverse bias conditions in the dual excitation setup. The field-dependent PL reduction was observed due to the dissociation of excitons under the applied electric field (Supplementary Fig. 7). The field-dependent quenching rate was obtained from PL quenching yield as $\left[PL(0) - PL(f)\right]/PL(0)$, where $PL(f)$ and $PL(0)$ are the PL intensities with and without an applied electric field, respectively. Figure 1 shows the experimental PL quenching yield along with the dissociation probabilities obtained from Onsager–Braun[27,28] (Supplementary Note 3). It is notable that with an increase in the electric field, the dissociation probability also increased as observed by the reduction of PL intensity in Supplementary Fig. 7c. Interestingly, the Onsager–Braun model shows a discrepancy with experimental data

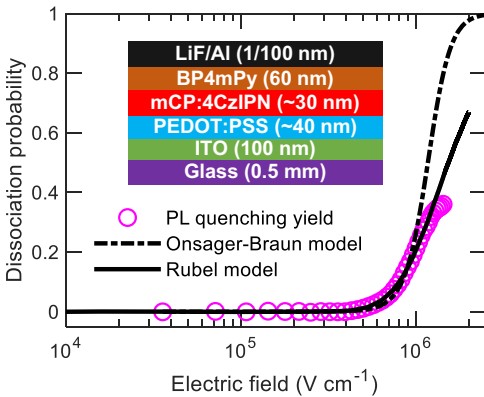

**Fig. 1 PL quenching in mCP:4CzIPN OLED under reverse bias.** Exciton dissociation probability as a function of the applied electric field. The dash-dot curve and solid line show the modeled quenching behavior as predicted by the Onsager–Braun model with $E_b$ = 0.56 meV, and Rubel model with $E_b$ = 0.48 meV, respectively. The inset shows the mCP:4CzIPN OLED device structure.

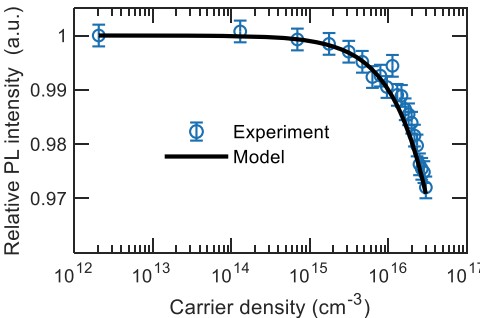

**Fig. 2 PL quenching in mCP:4CzIPN HOD.** The relative PL intensity from HOD under constant optical excitation as a function of carrier density. Fit to the quenching model is shown by the solid line.

beyond the applied field of ~$10^6$ V cm$^{-1}$, as the model curve becomes much steeper with a steplike increase. A slow increase in experimental dissociation probabilities indicates that the binding energy of the excitons might be affected by the environment[29]. To overcome this, we applied the Rubel model[29,30] (Supplementary Note 4) by taking account of the disorder effect. The calculated dissociation probability reproduced the experimental data well and the fitting parameters for the Rubel model are summarized in Supplementary Table 3. The extracted exciton binding energy ($E_b$) were 0.56 meV and 0.48 meV for Onsager–Braun and Rubel model, respectively. Field-induced quenching rate was calculated from the Rubel model by taking the value of $E_b$ = 0.48 meV along with parameters from Supplementary Table 3. To explore the effect of electric field-induced dissociation in singlet and triplet excitons, time-resolved PL responses from OLED are collected with dual excitation shown in Supplementary Fig. 8. Interestingly, PL decays collected under external electric field showed higher contribution from the delayed component compared to without electric field conditions. Thus, triplet quenching is assumed to be insignificant, and $R(f)T$ should not influence the dynamics of singlet and triplet densities studied in this work.

It is evident from Fig. 1 that electric field-induced exciton quenching is negligible at electric fields less than $5 \times 10^5$ V cm$^{-1}$. In HOD operation conditions under dual excitation, the maximum applied electric field was close to $3.4 \times 10^5$ V cm$^{-1}$. Therefore, the normalized PL drop in HOD at increased carrier densities as shown in Fig. 2 is instead predominantly due to

polaron-induced quenching. Fitting the experimental PL quenching data with Eqs. (1) and (2) yields $k_{SP}$ and $k_{TP}$ as $6.1 \times 10^{-12}$ cm$^3$ s$^{-1}$ and $5.8 \times 10^{-13}$ cm$^3$ s$^{-1}$, respectively.

The magnitude of SPA and TPA rates was further investigated in OLED configuration. For this investigation, we collected current density−voltage−luminance and EQE−current density properties of mCP:4CzIPN-based OLEDs (shown in Fig. 3 and summarized in Supplementary Table 4). The maximum EQE obtained from the OLEDs was $18 \pm 1\%$ at $J = 4.5$ mA cm$^{-2}$, indicating efficient charge recombination and exciton confinement in the OLEDs. Considering all the possible excited state pathways associated with the TADF process, the time evolution of singlet and triplet excited states in a TADF OLED can be modeled as

$$\frac{dP}{dt} = \frac{J(t)}{qd} - \gamma P^2, \qquad (3)$$

$$\frac{dS}{dt} = Y(J)\alpha\gamma P^2 - k_S S - k_{ISC}S + k_{RISC}T - k_{SP}SP \\ - k_{SS}S^2 - k_{ST}ST + \frac{\alpha}{2}k_{TT}T^2 - R(f)S, \qquad (4)$$

$$\frac{dT}{dt} = Y(J)(1-\alpha)\gamma P^2 - k_{RISC}T - k_T T + k_{ISC}S \\ - k_{TP}TP - \frac{1+\alpha}{2}k_{TT}T^2 - R(f)T, \qquad (5)$$

where $\gamma$ is the Langevin charge recombination rate expressed as $q(\mu_h + \mu_e)/(\varepsilon_r\varepsilon_0)$; $q$ is the elementary charge; $d$ is the width of the recombination zone assumed as 15 nm[14,19] in this work; $\varepsilon_0$ and $\varepsilon_r$ is the permittivity of free space and the relative permittivity, respectively; $\alpha$ is the singlet generation ratio and was assumed to be 0.25 in accordance with spin-statistics; $\mu_h$ ($\mu_e$) is the hole (electron) mobility of the emissive layer; $Y(J)$ is the charge balance factor[30] as function of $J$; $k_{SS}$, $k_{ST}$, and $k_{TT}$ refer to the rate constants of singlet-singlet annihilation (SSA), STA, and TTA, respectively. To quantify the SPA and TPA annihilation rates, EQE can be expressed as

$$\eta(J) = \eta_0 \frac{S(t = \infty, J)}{S_0}, \qquad (6)$$

where $\eta_0$ is the maximum experimental EQE, $S_0$ is the steady-state singlet density without SPA and TPA. In the OLED model, SPA and TPA were only accounted for holes as a previous report suggested quenching due to the electron being negligible in 4CzIPN[22]. The mCP electron mobility contributing to the Langevin recombination rate in Eq. (3) was neglected due to it being an order of magnitude smaller than that of the hole mobility[31]. The STA and TTA rate constants were taken from the literature as $1 \times 10^{-11}$ cm$^3$ s$^{-1}$ and $5 \times 10^{-18}$ cm$^3$ s$^{-1}$, respectively[18], and used to fit the EQE curve. Additionally, SSA was not considered in our calculations as a recent report showed a minor contribution of SSA under electrical excitation for TADF OLEDs[17]. The experimental EQE, calculated fit, and extracted charge balance factor as a function of current density using Eqs. (3)–(6) are shown in Fig. 3b. The best fit was obtained with $k_{SP}$ and $k_{TP}$ as $2 \times 10^{-11}$ cm$^3$ s$^{-1}$ and $9.1 \times 10^{-13}$ cm$^3$ s$^{-1}$, respectively. Interestingly, both the SPA and TPA rates from OLEDs were slightly higher than the value obtained from dual excitation method in HODs. This hints toward the existence of additional loss mechanisms in OLEDs that may potentially also induce EQE roll-off, such as optical loss mechanisms[32,33] and joule heating[34].

To obtain more insights on the order of magnitude of $k_{SP}$ and $k_{TP}$ in typical TADF emitters, we have studied another well-known green-emitting TADF compound, 3-(9,9-dimethylacridin-10(9H)-yl)-9H-xanthen-9-one (ACRXTN). By adopting the same

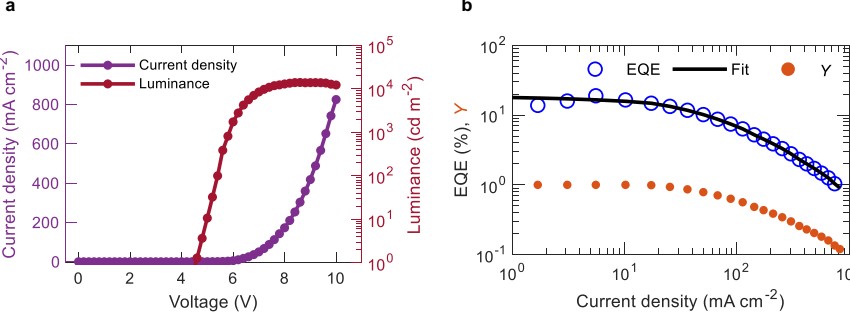

**Fig. 3 Device characteristics of mCP:4CzIPN OLEDs. a** Current density−voltage−luminance plot. **b** EQE−current density plot. The solid line represents fit obtained from Eqs. (3)–(6).

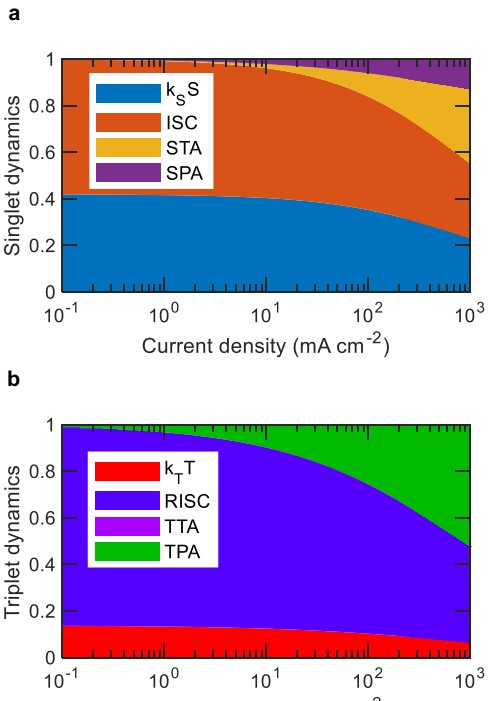

**Fig. 4 Relative contributions of the different excitonic processes associated with TADF emission.** Relative contributions are plotted for mCP:4CzIPN OLEDs as a function of current density. **a** Deactivation of singlet excited state. **b** Deactivation of triplet excited state.

methods applied on 4CzIPN, investigation with non-doped ACRXTN HODs (details in Supplementary Figs. 9–11, and Supplementary Table 5) and OLEDs (details in Supplementary Fig. 12 and Supplementary Table 6) was performed. The $k_{SP}$ and $k_{TP}$ found from non-doped ACRXTN HOD were $1.3 \times 10^{-12}\ \mathrm{cm^3\ s^{-1}}$ and $5.4 \times 10^{-13}\ \mathrm{cm^3\ s^{-1}}$, respectively and from OLED device were $4 \times 10^{-12}\ \mathrm{cm^3\ s^{-1}}$ and $6 \times 10^{-13}\ \mathrm{cm^3\ s^{-1}}$, respectively. Similar to 4CzIPN, higher $k_{SP}$ and $k_{TP}$ values were obtained for ACRXTN-based OLED compared to HOD. This further substantiates the existence of the additional loss mechanisms in OLEDs. However, taking account of loss mechanisms such as joule heating can be complicated due to the temperature dependence of the RISC rate[8].

## Discussion

To study the interplay between different deactivation pathways in mCP:4CzIPN OLEDs, the relative contribution of all the relevant mechanisms for singlet and triplet excited states are plotted in

Fig. 4 using Equations (3)–(5) applying the extracted parameters from the OLEDs, while assuming a perfectly charge-balanced device as $Y(J) = 1$. At lower current densities the dominating deactivation processes for singlet and triplet states are ISC and RISC, respectively, suggesting a large number of excitons remain either in singlet or triplet states as part of the spin cycling process. With increasing current density, the area corresponding to STA, SPA, and TPA gradually increases until these loss mechanisms start to dominate over RISC, ISC, fluorescence ($k_S S$), and phosphorescence ($k_T T$) pathways. From singlet exciton dynamics, it is clear that both STA and SPA can significantly contribute to the density-driven singlet loss mechanism under high bias conditions. Yet the impact of STA is almost three times higher than SPA observed at high bias conditions (e.g., $10^3\ \mathrm{mA\ cm^{-2}}$), which emphasizes the necessity of reducing the $k_{ST}$ value to improve EQE roll-off in TADF OLEDs. Interestingly, at $10^3\ \mathrm{mA\ cm^{-2}}$, only ~21% of total generated singlet excitons participate in the fluorescence pathway to emit light. In the case of triplet dynamics, the relative contribution from TTA annihilation is negligible compared to TPA as it is invisible in Fig. 4b, which is due to the low $k_{TT}$ of $5 \times 10^{-18}\ \mathrm{cm^3\ s^{-1}}$. Though materials with higher TTA rates can become the dominant deactivation pathways for triplets. Approximately 52% of generated triplet excitons are deactivated at $10^3\ \mathrm{mA\ cm^{-2}}$ due to TPA, which is expected to greatly contribute to device degradation. It is important to note that STA and SPA can also contribute to the device degradation process. However, owing to the spin-statistics and comparably shorter singlet lifetimes than triplet lifetimes, the singlet population is a few orders of magnitude lower than the triplet population under steady-state conditions (Supplementary Fig. 13). Therefore, the absolute number of triplet excitons deactivated by TPA is higher than SPA or STA, which sets TPA as the major contributor to the device degradation across the different annihilation processes. Additionally, TPA has the potential to indirectly reduce TADF OLED efficiencies by reducing the steady-state singlet population as it deactivates the triplets such way that fewer number of triplets will become available for upconversion to singlets via RISC.

However, it is unclear from Fig. 4 what the impact of TPA is on the singlet population. To put both polaron-induced quenching processes into the same picture and relative to singlet dynamics, we have taken a parameter as $\delta = (S_{SP} - S_{TP})/S_0$ where $S_{SP}$ and $S_{TP}$ denotes the steady-state singlet density from Eqs. (4) and (5) with SPA while TPA is ignored ($k_{TP}TP = 0$), and TPA while SPA is ignored ($k_{SP}SP = 0$), respectively; $S_0$ has the same description as in Eq. (6). It is notable that if $\delta$ is negative ($S_{SP} < S_{TP}$), then singlet loss due to SPA dominates over TPA. Conversely, if $\delta$ is positive ($S_{SP} > S_{TP}$), then singlet loss from TPA will dominate over SPA. There is also the possibility of $\delta = 0$, in which SPA and TPA separately result in the same singlet density ($S_{SP} = S_{TP}$), implying an equal amount of loss caused by the processes

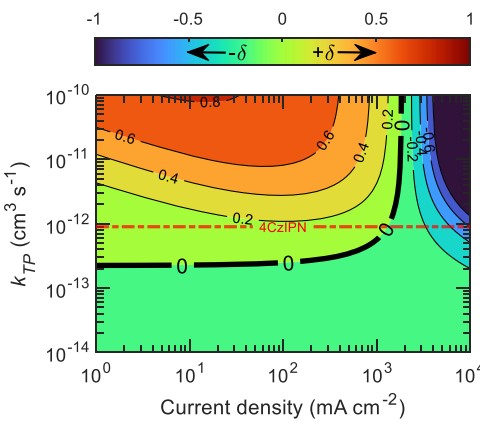

**Fig. 5 Disentangling the impact of TPA and SPA.** $\delta$ versus current density, as a function of $k_{TP}$ values for mCP:4CzIPN OLEDs. In the plot, dominating loss mechanism to singlet density via SPA (negative regions) or indirectly TPA (positive regions). The line '0' represents the contour where the loss due to SPA and TPA individually, results in the same singlet density. The dash-dot straight line represents experimentally obtained $k_{TP}$ as $9.1 \times 10^{-13}$ cm$^3$ s$^{-1}$ for 4CzIPN OLED.

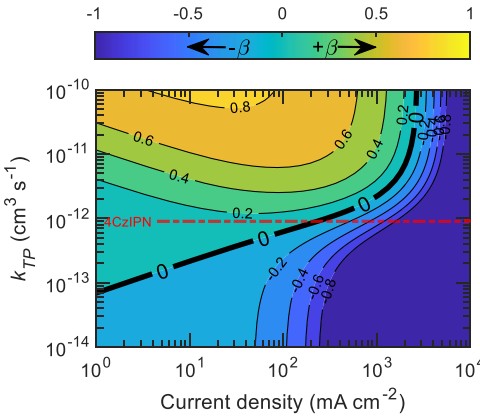

**Fig. 6 Disentangling the impact of STA and TPA.** $\beta$ versus current density, as a function of $k_{TP}$ values for mCP:4CzIPN OLEDs. In the plot, dominating loss mechanism to singlet density via STA (negative regions) or indirectly TPA (positive regions). The line '0' represents the contour where the loss due to STA and TPA individually results in the same singlet density. The dash-dot straight line represents experimentally obtained $k_{TP}$ as $9.1 \times 10^{-13}$ cm$^3$ s$^{-1}$ for 4CzIPN.

individually. Figure 5 shows calculated $\delta$ using Eqs. (3)–(5) at different current densities over a range of TPA rate constants (varied by 5 orders of magnitude), while ignoring the charge balance factor and keeping all the other parameters constant. The dash-dot horizontal straight line represents the intrinsic $k_{TP}$ of 4CzIPN OLED as previously calculated. It is evident from the position of this line that TPA mostly dominates over SPA in current densities lower than $\sim 2 \times 10^3$ mA cm$^{-2}$, as it remains in the positive $\delta$ regime. However, with the further increase in current density, the effect of SPA starts to dominate over TPA. For a different TADF material, the interplay between different annihilation processes may modify the relative position $k_{TP}$ such that either SPA or TPA can dominate over a certain range of current density. A similar comparison can be made between STA and TPA by taking $\beta = (S_{ST} - S_{TP})/S_0$, where $S_{ST}$ represents steady-state singlet density from Eqs. (4) and (5) with STA while TPA is ignored. Note that positive and negative $\beta$ represent the

dominance of TPA and STA, respectively. The values of $\beta$ are plotted in Fig. 6 as a function of current density and TPA rate constants. From the intrinsic $k_{TP}$ of 4CzIPN (dash-dot horizontal straight line) and '0' contour line, it is clear that the loss due to TPA dominates over STA under current densities lower than $\sim 10^2$ mA cm$^{-2}$, while for higher current densities ($>10^2$ mA cm$^{-2}$) STA starts to dominate over TPA. As rapid EQE roll-off was observed in the steady-state device operation beyond the current density of $\sim 10^2$ mA cm$^{-2}$, it is thus appropriate to infer that the contribution of STA in EQE roll-off is higher than that of TPA. Importantly, SPA and STA processes predominantly take place by long-range Förster energy transfer and are dependent on the overlap between singlet emission and polaron absorption spectra, and triplet absorption spectra, respectively. Conversely, TPA is limited to short-range Dexter energy transfer and mainly depends on triplet diffusion length. Reducing Förster radius may reduce $k_{SP}$ and $k_{ST}$, while lower triplet diffusivity is expected to reduce $k_{TP}$. This way of disentangling the impact of SPA, STA, and TPA processes in TADF OLEDs can be helpful in order to determine where most effort should be invested in order to reduce annihilation based upon the dominance of SPA, STA, or TPA over a certain range of current densities.

Finally, we investigated two extreme conditions when SPA and TPA become more dominant than the natural fluorescence process. We have defined a current density $J_{SP}$ when $k_S + k_{ISC} \le k_{SP}P$ and $J_{TP}$ when $k_T + k_{RISC} \le k_{TP}P$, which represents SPA and TPA as the main pathway of deactivation, respectively. By assuming $Y(J) = 1$, $S \approx \tau_P \alpha \frac{J}{ed}$, and $T \approx \tau_D(1 - \alpha)\frac{J}{ed}$, from Eqs. (3)–(5), we can get

$$J_{SP} \approx \frac{ed\gamma}{k_{SP}^2 \tau_P^2 \alpha}, \quad J_{TP} \approx \frac{ed\gamma}{k_{TP}^2 \tau_D^2 (1 - \alpha)} \qquad (7)$$

It is clear from Eq. (7) which parameters should be optimized to reduce the effect of SPA and TPA in TADF OLEDs. The value of $J_{SP}$ can be maximized by reducing $\tau_P$, $k_{SP}$, and $\alpha$, while the value of $J_{TP}$ can be maximized by reducing $\tau_D$ and $k_{TP}$, and increasing $\alpha$. However, as the delayed lifetime is inversely proportional to the delayed efficiency and rate of reverse intersystem crossing ($\tau_D = 1/\Phi_D k_{RISC}$, considering $k_T = 0$), it can be assumed that increasing $k_{RISC}$ and $\Phi_D$ are desirable to reduce the effect of TPA. The impact of both SPA and TPA can be reduced by broadening of recombination zone and increasing the Langevin recombination rate constant. Previous reports have shown how increasing the width of the recombination zone can significantly reduce EQE roll-off in phosphorescent OLEDs[14]. In addition, $\gamma$ can be increased by maximizing $(\mu_h + \mu_e)$, which can be obtained by choosing appropriate materials with high mobilities. It is noteworthy that not only high but also balanced hole and electron mobilities are beneficial to balance charge fluxes and broaden the recombination zone[35]. We have varied $\gamma$ for several orders of magnitude and calculated EQE using Eqs. (3)–(6) keeping other parameters fixed with the measured values to further evaluate the role of $\gamma$ in the EQE roll-off process. Figure 7a shows a significant reduction in EQE roll-off with an increase in $\gamma$ value. A critical current density ($J_{50}$) at which EQE drops to 50% to its maximum value is plotted for each gamma value at Fig. 7b. For $\gamma$ below $10^{-12}$ cm$^3$ s$^{-1}$, $J_{50}$ does not change noticeably as it is limited by the polaron-induced quenching. If $\gamma$ is increased by an order to $10^{-11}$ cm$^3$ s$^{-1}$, the critical current density increases from $J_{50} = 30$ mA cm$^{-2}$ to $J_{50} = 117$ mA cm$^{-2}$. A further increase of the $\gamma$ results in a significantly high $J_{50}$ value, which indicates any slight improvement in the recombination in this order of magnitude will significantly reduce efficiency roll-off. However, there may be a trade-off associated with achieving high orders of $\gamma$ values through increasing carrier mobilities as it may also increase the triplet diffusivity, which may give rise to $k_{TP}$ and $k_{TT}$[36,37].

**a**

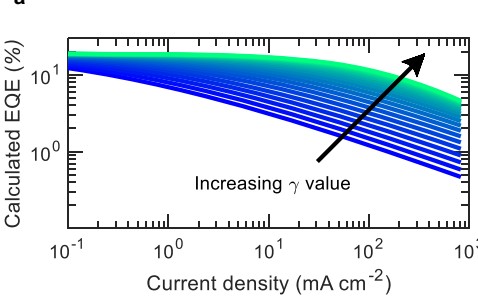

**b**

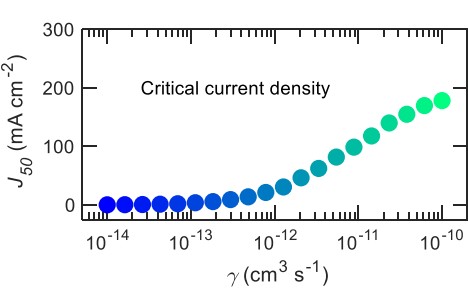

**Fig. 7 Impact of γ in EQE roll-off. a** Calculated EQE for different γ values. **b** Critical current density $J_{50}$ for each γ value.

This negative effect of high γ can, in part be mitigated by spatially separating emitter molecules through low doping concentration or designing molecules with increased size.

In summary, we have determined polaron and field-induced quenching parameters independently in 4CzIPN-based TADF OLEDs. Using experimentally quantified parameters and theoretical models, EQE roll-off was investigated. After performing the analysis on the interplay of different intrinsic processes, it is evident that polaron-induced quenching is one of the contributors to the efficiency roll-off in TADF OLEDs and device degradation. Charge imbalance and STA were found to be the main contributors to the efficiency roll-off in the 4CzIPN-based OLEDs. However, the impact of field-induced quenching is negligible under the conventional OLED operating regime. Disentangling losses due to long-range SPA, STA, and short-range TPA separately may provide guidelines for the selection of materials and offer a route to design OLEDs with reduced efficiency roll-off and device degradation. We have also shown that maximizing the charge recombination rate may lead to OLEDs in which $J_{50}$ values can be significantly increased due to the reduction of the impact of polaron-induced quenching. Our simulated dynamics not only explain the efficiency roll-off in TADF OLEDs but also provide effective strategies for device optimization.

## Methods

**Photophysical measurements**. Thin-film samples for photophysical studies were prepared by spin-coating the mCP:5 wt% 4CzIPN blend from 20 mg mL$^{-1}$ in chloroform solution at 1500 rpm on quartz substrates to give a film thickness of ~250 nm. UV-visible absorption and PL spectra were measured using a Cary-5000 UV-Vis spectrophotometer and FS5 Spectrofluorometer, respectively. Time-correlated single-photon counting (TCSPC) measurements were performed with a Jobin-Yvon Fluorolog-3, by exciting the samples at an excitation wavelength of 372 nm generated by a pulsed LED and an instrument response of about 1 ns. To minimize the impact of oxygen quenching, samples were kept under the vacuum (~10$^{-5}$ mbar). The thin film PL quantum yield was measured in an integrating sphere under continuous nitrogen flow[38].

**Electrical characterization**. The OLEDs and HODs were fabricated by using glass substrates with 100 nm thick pre-patterned indium tin oxide (ITO) as the anode. The substrates were cleaned ultrasonically with Alconox solution, deionized water, acetone, and 2-propanol for 10 min in each solvent. After that, the substrates were treated with UV-ozone for 30 min. Poly(3,4-ethylenedioxythiophene):poly(styrenesulfonate) (PEDOT:PSS aqueous solution−Ossila AI 4083) was then spin-

coated at 4000 rpm onto the substrates and annealed at 15 °C for 20 min. For OLED fabrication, the solution of host and emitter were prepared separately in chloroform and mixed to produce the mCP:5 wt% 4CzIPN blend solution with a concentration of 5 mg mL$^{-1}$. This mixed solution was spin-coated onto the PEDOT:PSS layer at 1500 rpm for 1 min in the N$_2$ atmosphere. The thickness of the spin-coated layers was determined by Dektak 150 profilometer. Finally, 3,3′,5,5′-tetra[(m-pyridyl)-phen-3-yl]biphenyl (BP4mPy), LiF, and Al were deposited via thermal evaporation under vacuum (~10$^{-7}$ mbar). The detailed OLED fabrication procedure for non-doped ACRXTN can be found in the literature[17]. All the steady-state current density−voltage−luminance characteristics of OLEDs were measured using the Keithley 2400 source meters, a calibrated photomultiplier tube (PMT, Hamamatsu H10721-20), and a luminance meter (Konica Minolta LS100). The external quantum efficiency (EQE) was calculated (assuming Lambertian emission) using the brightness, current density, and emission spectrum of the device[39]. For HODs, the blend solution of mCP:4CzIPN and non-doped ACRXTN with a concentration of 20 mg mL$^{-1}$ was used to produce the active layer. The current density−voltage response with HODs was measured using an Agilent B1500A semiconductor device analyzer.

**Polaron- and field-induced quenching measurements**. For quenching experiments, HODs/OLEDs were simultaneously excited optically with a HeCd laser with an excitation wavelength of 325 nm and electrically with a Keithley 2400 source meter (Supplementary Fig. 2). The laser spot size was controlled by an iris diaphragm. The voltage-dependent PL response from HODs/OLEDs was collected using optical fiber and spectrometer (Hamamatsu, Mini-spectrometer TM series, C10083CA) with a spectral resolution of 5 nm. An appropriate cutoff filter was used to remove the excitation signal from the PL response. Time-resolved measurements were performed by optically exciting the OLEDs at an excitation wavelength of 337 nm with a 3.5 ns pulse using the randomly polarized nitrogen-gas laser (Stanford Instruments, NL-100) operating at 20 Hz. Time-resolved PL from devices was collected using a PMT (Hamamatsu H10721-20) with a response time of 0.57 ns connected to a digital oscilloscope (Teledyne LeCroy, 2 GHz).

## Data availability

The data that support the findings of this study are available from the authors upon request.

## Code availability

The code used for the data fitting and simulations used in this study is available from the authors upon request.

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

## Acknowledgements

We are grateful to the Australian Research Council (ARC DP200103036), Department of Industry, Innovation and Science (AISRF53765), and JSPS Core-to-Core Program (grant number: JPJSCCA20180005) for financial support. We acknowledge Timofey Golubev from Michigan State University for helping us with the drift-diffusion model. M.H. was supported by the Australian Government's Australian Postgraduate Award (APA) and S.S. was supported by the UQ Research and Training Program. This work was performed in part at the Queensland node of the Australian National Fabrication Facility Queensland Node (ANFF-Q)–a company established under the National Collaborative Research Infrastructure Strategy to provide nano- and micro-fabrication facilities for Australia's researchers.

## Author contributions

M.H. and E.B.N. conceived the idea of this work. M.H fabricated and characterized the HODs and OLEDs. M.H performed photophysical studies, theoretical calculations, and device simulation. F.B performed field-induced quenching simulations. M.H., S.S., A.S., and J.S. designed and built the dual excitation setup for quenching measurements. C.A. provided the materials. S.K.M.M. purified the materials. M.H., E.B.N., and S.-C.L. drafted the manuscript. All the authors contributed to data analysis and discussion of the results. E.B.N., S.-C.L., and C.A. initiated and supervised the project.

## Competing interests

The authors declare no conflict of interests.
