## [Peer Review File · Nature Communications]

Probing polaron-induced exciton quenching in TADF-based organic light-emitting diodesREVIEWER COMMENTS

Reviewer #1 (Remarks to the Author):

This work describes the mechanism of roll-off and device degradation related to polaron-induced exciton quenching, namely singlet-polaron annihilation (SPA) and triplet-polaron annihilation (TPA) in TADF-based OLEDs. The work has been done meticulously and, for the most part, described clearly, with detailed device data and solid photophysical data support. However, there are some points listed below that need further clarification; therefore, I recommend publication of the manuscript after minor revision.

1) For Figure 5, the authors claimed that SPA and TPA were the major factors that contribute to the device's efficiency roll-off and degradation. However, from Figure 5a, it seems that STA would be a major factor that contributes to the singlet dynamics when the voltage is larger than 6 V. The authors mentioned "the impact of STA is almost five times higher than SPA...", which emphasized the necessity of reduction of k_{ST} in the original text. Later in the text, the authors seem to overlook the significance of STA, and to assign further contribution to roll-off effects to SPA rather than STA. It would be convincing to the reader if the authors can explain the boost in STA at higher voltage and interpret the role of STA in the singlet dynamics.

2) The formation of polaron would influence the carrier mobility in the system. The authors haven't clarified the role of carrier mobility in the device optimization.

3) There are many fittings in this work, both the main manuscript and supplementary information, but none of the fitting parameters has been given to support the goodnesses of fit. The authors should give all the fitting parameters to convince the reader about the correctness of all fittings.

4) The authors introduced a hole-only device consisting of a blend of 4CzIPN: mCP, due to negligible electron-transporting property in 4CzIPN, for investigating the effect of polaron-induced quenching. However, there is no clear information on energy levels related to the blend structure to confirm that mCP is the dominant bipolar charge carrier transporter and then 4CzIPN as the dopant through energy transfer -- or that 4CzIPN also contributes to the hole transport directly working as a hole trapper.

5) The authors claim that the PL intensity drop with increase in voltage can be attributed to the SPA, TPA, and field-induced quenching processes. But in Supplementary Fig. 3, there is an apparent negligible PL intensity drop observed with an increase in applied voltages, which might suggest that the effect of SPA, TPA, and field-induced quenching processes were less serious than what the main manuscript discussion emphasizes.

6) Theoretically, the PL spectra shown in Supplementary Fig. 3 and 4 should be the same due to the lack of EL emission observed in the HODs. A shoulder appears at around 550 nm in HODs but not in 4CzIPN: mCP blend. One would expect the same emission profile in both the HODs and OLED film consisting of 4CzIPN:mCP, whereas the (albeit tiny) shoulder that occurs at around 550 nm only appears in the former.

7) In Supplementary Table 1, ϕ_P has been denoted as ϕ_D -- although ϕ_P and ϕ_D were 41.94% and 42.05%, respectively. However, these values are presented as 41% and 43%, respectively, instead in the main paper.

8) Equations inserted in the main paper should be numbered in order and referred to in the text.

Reviewer #2 (Remarks to the Author):

The authors studied the effect of polarons-induced exciton quenching to understand the efficiency roll-off in TADF based OLEDs. To investigate SPA and TPA, drift-diffusion numerical modeling, exciton dissociation theory, kinetic exciton dynamics were introduced and successfully allowed

understanding of bimolecular exciton kinetics.

The manuscript was well organized and the results of experiment and modeling were convincing.

However, some issues should be addressed before the publication.

1. In drift-diffusion modeling for determining the polaron density (actually field-dependent mobility in the modeling), was the trap-dynamics included? Active layer in the HOD included both mCP host and 4CzIPN dopant and hole trap was existed considering the HOMO level.

Is there a reason to set the DOS as fitting parameter? I wonder the effect of DOS value to the result of the simulation.

2. In EQE roll-off fitting (Fig.4), the charge balance factor (γ) was decreased from 1 to 0.1 with increasing the current density. I think this change was quite large. Considering the factor, recombination zone would be changed with driving voltage. How was the EL spectrum shift dependent on the voltage?

3. The calculation results in Fig. 5 demonstrated that TPA was main contributor of the roll-off and TTA showed very low portion in triplet dynamics. I think it might be attributed that TTA coefficient was adopted from the literature and fixed and TPA was set as fitting parameter in simulation.

4. Were the simulations in the paper commercial program or in-house code? Brief information of the simulation should be included in the paper such as programming language and fitting algorithm for the readers.

5. I suggest that x-axis of fig.5 would be appropriate to be the current density or luminance for intuitive understanding since the light emission did not occurred in low voltage.

6. The molecular structure of BP4mPy was not included. Also, inclusion of device structure with frontier orbital energetics would be good to better understanding of the readers.

7. Some formats should be modified. Number of equations were wrong. The symbols of PLQY for prompt and delay in Si Table 1d were same. The structure of SI should be modified in terms of the sections or individual figures and tables. In terms of the figures and tables, all components should be addressed in the main manuscript.

Reviewer #3 (Remarks to the Author):

Presented by Hasan et al., this paper managed to discuss polaron-induced exciton quenching in TADF-based organic light-emitting diodes (OLEDs). By utilizing steady-state photoluminescence (PL) and electroluminescence (EL) measurements, the authors attempted to quantify the

quenching rates of singlet-polaron annihilation and triplet-polaron annihilation then to analyze their influence on the OLEDs performance. As understanding quenching mechanisms in OLEDs plays a crucial role in improving device efficiencies, this paper is of importance to the materials science community, thus qualified for being published in Nat. Commun. Nevertheless, several issues should be addressed properly before further consideration for being published. Comments and suggestions are listed below:

1. Apart from exciton-polaron annihilation, the quenching possibly arises from local heat or even from the decomposition of emitters under a high voltage. The authors should at least provide direct experimental evidence of polarons and comment on the influence of excitation (or current) density and lattice heating effect on their measurements.
2. The authors should comment on why a dual excitation setup was specifically used for this study.
3. In Supplementary Fig.4b, the prompt and delay lifetimes are obtained through tail fit instead of deconvolution fit. Since the decay profile (purple dots) is already a convolution between excitation source and population decay, it is recommended to perform deconvolution fit with instrumental response function to obtain an accurate prompt decay time, Φ_P and Φ_D .
4. The parameters, k_{SS} , k_{SP} , and k_{TP} , are rate constants, which should be investigated by time-resolved measurements (PL and/or EL) instead of merely by the fitting results from steady-state approaches.
5. I recommend the authors to move Fig. 1 to the supplementary information, as it appears that the purpose of simulating current density-voltage response is only to obtain the hole mobility and characteristic field value, both of which are not further evaluated.
6. What is the base for the authors to attribute the field dependent PL reduction solely to the dissociation of singlet excitons without considering the dissociation of triplet excitons? The excited-state population of the singlet and triplet excitons should be roughly at the same level, since the prompt and delayed efficiency, which corresponds to the singlet and triplet excitons, respectively, share a similar contribution to the QY.
7. In main text, P.8, Eq. (3), the recombination zone (d) is set as 15 nm. A reference is required for this assumption.
8. In main text, P.8, Eq. (4), a singlet generation ratio of 0.25 is assigned for both exciton generation and triplet-triplet annihilation based on spin statistics. However, according to previous TTA work on anthracene (J. Am. Chem. Soc. 1981, 103, 7159-7164), it is easily shown that the possibility of excited singlet pair generated from triplet pairs will not exceed a factor of 1/9. Thus the authors should reevaluate the coefficient used for k_{TT} . Furthermore, the k_{TT} term should be a plus sign as it stands for singlet generation. Likewise, in Eq. (5), why is the coefficient of k_{TT} being $-(1+\alpha)$ instead of -1 ? (Phys. Rev. B 2002, 66, 035321)
9. The fitted k_{SP} and k_{TP} from PL ($6.1 \times 10^{-12} \text{ cm}^3\text{s}^{-1}$ and $5.8 \times 10^{-13} \text{ cm}^3\text{s}^{-1}$, respectively) and EL measurements ($2 \times 10^{-11} \text{ cm}^3\text{s}^{-1}$ and $9.1 \times 10^{-13} \text{ cm}^3\text{s}^{-1}$, respectively) for mCP:4CzIPN (also for ACRXTN OLED) is not reasonable to me. As the lifetime of singlet excitons is known to be much faster than that of triplet excitons, even disregarding the population gap between singlet and triplet excitons, the chance for polarons to annihilate singlet excitons should be much lower than to annihilate triplet excitons. How do the authors explain the results that $k_{SP} > k_{TP}$?
10. For the fitting processes to acquire k_{SP} and k_{TP} , the authors state that the STA and TTA rate constants were taken from the reference paper (Appl. Phys. Lett. 113, 083301 (2018)). Yet, this reference paper reported a series of simulations with different sets of STA and TTA values. Which set of values did the authors adopted in this manuscript? Please specify all the fixed values in the fit process at least in the supplementary information.

Minor points:

1. All equations should be properly labeled from Eq.(1) to Eq.(7).

Reviewers' comments:

Reviewer #1 (Remarks to the Author):

This work describes the mechanism of roll-off and device degradation related to polaroninduced exciton quenching, namely singlet-polaron annihilation (SPA) and triplet-polaron annihilation (TPA) in TADF-based OLEDs. The work has been done meticulously and, for the most part, described clearly, with detailed device data and solid photophysical data support. However, there are some points listed below that need further clarification; therefore, I recommend publication of the manuscript after minor revision.

Response: We thank the Reviewer for the comments and careful evaluation of our work. We are delighted and grateful for the Reviewer's positive feedback.

1) For Figure 5, the authors claimed that SPA and TPA were the major factors that contribute to the device's efficiency roll-off and degradation. However, from Figure 5a, it seems that STA would be a major factor that contributes to the singlet dynamics when the voltage is larger than 6 V. The authors mentioned "the impact of STA is almost five time higher than SPA..., which emphasized the necessity of reduction of kST" in the original text. Later in the text, the authors seems to overlook the significance of STA, and to assign further contribution to roll-off effects to SPA rather than STA. It would be convincing to the reader if the authors can explain the boost in STA at higher voltage and interpret the role of STA in the singlet dynamics.

Response: We thank the Reviewer for pointing out the importance of STA in the singlet dynamics and EQE roll-off in TADF OLEDs. Indeed, both SPA and STA directly reduces singlet density. From Fig. 5a (now Fig. 4a in the revised manuscript), it is clear that STA mostly dominates over SPA as the deactivation pathway. However, any comparison between the impact of STA and TPA had not been discussed in the manuscript. Hence, to further highlight the role of the STA process, we have now added a new part in the manuscript.

Added part:

Fig. 6. Disentangling the impact of STA and TPA. β versus current density, as a function of k_{TP} values for mCP:4CzIPN OLEDs. In the plot, dominating loss mechanism to singlet density *via* STA (negative regions) or indirectly TPA (positive regions). The line ‘0’ represents the contour where the loss due to STA and TPA individually results in same singlet density. The dash-dot line represents experimentally obtained k_{TP} as $9.1 \times 10^{-13} \text{ cm}^3 \text{ s}^{-1}$ for 4CzIPN.

“A similar comparison can be made between STA and TPA by taking $\beta = (S_{ST} - S_{TP})/S_0$, where S_{ST} represents steady-state singlet density from Eqs. (4)–(5) with STA while TPA is ignored. Note that positive and negative β represents dominance of TPA and STA, respectively. The values of β are plotted in Fig. 6 as a function of current density and TPA rate constants. From the intrinsic k_{TP} of 4CzIPN (dash-dot horizontal straight line) and ‘0’ contour line, it is clear that the loss due to TPA dominates over STA under current densities lower than $\sim 10^2 \text{ mA cm}^{-2}$, while for higher current densities ($> 10^2 \text{ mA cm}^{-2}$) STA starts to dominate over TPA. As rapid EQE roll-off was observed in the steady-state device operation beyond the current density of $\sim 10^2 \text{ mA cm}^{-2}$, thus it is appropriate to infer that the contribution of STA in EQE roll-off is higher than that of TPA”.

Added line in the Conclusion

“Charge imbalance and STA were found to be the main contributors of the efficiency roll-off in the 4CzIPN-based OLEDs”.

2) The formation of polaron would influence the carrier mobility in the system. The authors haven't clarified the role of carrier mobility in the device optimization.

Response: We appreciate the Reviewer's comment. Both electric field and polaron density influence the carrier mobility in the device. In organic semiconductors, increase in carrier mobility can increase charge recombination to form excitons. However, orders of magnitude difference in electron and hole mobility complicate the device optimization process to obtain the maximum charge balance factor. Many attempts have been made to improve the electron drift mobility. However, it is still not comparable with hole mobility [*Chem. Mater.* **16**, 4556-4573 (2004); *Nat. Mater.* **15**, 628-633 (2016)]. A balanced charge transport which is essential for optimum device performances, can be achieved by either increasing the electron mobility or decreasing the hole mobility [*Nat. Mater.* **15**, 1120-1127 (2016)].

3) There are many fittings in this work, both the main manuscript and supplementary information, but none of the fitting parameters has been given to support the goodnesses of fit. The authors should give all the fitting parameters to convince the reader about the correctness of all fittings.

Response: We acknowledge the Reviewer's comment. To check accuracy of our data fitting, we have applied the root-mean square error (RMSE) method. The calculated RMSE for all the fitting in this work was <1, indicating that overall good agreement with experimental data was achieved.

4) The authors introduced a hole-only device consisting of a blend of 4CzIPN: mCP, due to negligible electron-transporting property in 4CzIPN, for investigating the effect of polaroninduced quenching. However, there is no clear information on energy levels related to the blend structure to confirm that mCP is the dominant bipolar charge carrier transporter and then 4CzIPN as the dopant through energy transfer -- or that 4CzIPN also contributes to the hole transport directly working as a hole trapper.

Response: We appreciate the Reviewer's comment. To incorporate more information about charge injection/transport/exciton formation process in HOD/OLED, we have now added their energy level diagrams in the Supplementary Information (Supplementary Fig. 2, Fig. 7, Fig. 10, and Fig. 11).

Since the dopant molecule 4CzIPN has a HOMO level comparable to that of the host mCP and lightly doped EML (mCP:5 wt% 4CzIPN) suggests that the charge injection/transport process in the HODs might be dominated by mCP [*Adv. Funct. Mater.* **19**, 3157-3164 (2009); *Appl. Phys. Lett.* **83**, 3818-3820 (2003)]. Figure below shows the current density–voltage response from the HOD and EOD devices along with their energy diagrams. It can be assumed from current density–voltage response that doped EML is more favourable for hole injection/transport rather than electron injection/transport. Furthermore, the electron transport can be slightly contributed by 4CzIPN due to its deeper LUMO (more negative) than mCP. For the above-mentioned reasons, we believe charge transport in the OLEDs was mainly dominated by holes.

Figure | HOD and EOD device characteristics with 4CzIPN. **a** Energy level diagrams of the materials employed in the 4CzIPN HOD. **b** Energy level diagrams of the materials employed in the 4CzIPN EOD. **c** Experimental current density as function of applied voltage for HOD (red) and EOD (blue).

5) The authors claim that the PL intensity drop with increase in voltage can be attributed to the SPA, TPA, and field-induced quenching processes. But in Supplementary Fig. 3, there is an apparent negligible PL intensity drop observed with an increase in applied voltages, which might suggest that the effect of SPA, TPA, and field-induced quenching processes were less serious than what the main manuscript discussion emphasizes.

Response: We thank the Reviewer for raising a valid point. The exciton generation in the polaron-induced quenching experiment was carried out with a low power optical of $\sim 50\mu\text{W}$ excited at 325 nm, which is approximately equivalent to $\sim 8.2 \times 10^{13}$ photons per second. By considering photon absorption from different interlayers and the absorption of mCP at 325 nm, the actual number of generated excitons in the active layer might be even lower than the calculated photon numbers. Since the amount of loss due to annihilation process is exciton density dependent, in our experiment the PL intensity drop due to SPA and TPA processes was expected to be low considering the low photon density. In addition, the HOD devices were operated under low current densities to avoid any device breakdown, which provided low polaron density as well as low PL quenching. Though we believe the PL quenching was still significant enough to extract polaron-induced quenching rate. A similar experimental setup has been previously reported for phosphorescent HODs in *Adv. Funct. Mater.* **24**, 6074-6080 (2014), where only 3% PL intensity drop due to polaron quenching was observed and later utilised to extract TPA rate constant. However, it is important to mention that the number of exciton and polaron generation in OLEDs *via* electrical injection at high current densities can exceed these numbers by several orders of magnitude. As a result, the impact of this quenching can significantly increase under those circumstances.

6) Theoretically, the PL spectra shown in Supplementary Fig. 3 and 4 should be the same due to the lack of EL emission observed in the HODs. A shoulder appears at around 550 nm in HODs but not in 4CzIPN: mCP blend. One would expect the same emission profile in both the HODs and OLED film consisting of 4CzIPN:mCP, whereas the (albeit tiny) shoulder that occurs at around 550 nm only appears in the former.

Response: We thank the Reviewer for pointing out the deviation in the collected PL spectra from the HOD devices and quartz substrates. Figure below shows the normalized voltage dependent PL spectra collected from the HOD. Since there is no variation in the voltage

dependent PL in Fig. a, the possibility of spectral change due to polaron absorption can be abandoned. In Fig. b PL spectra collected at 0 V bias (no possibility of EL emission) and PL spectra from quartz substrates are plotted. There is a small variation in the PL spectra observed, which we believe can be due to the cavity effect in the HOD. [*Chem. Phys. Lett.*, **644**, 62-67 (2016)].

Figure | PL comparison a Normalized voltage dependent PL spectra collected from HOD. b Normalized PL spectra from HOD without applied bias voltage and PL spectra from quartz substrate.

7) In Supplementary Table 1, ϕ_P has been denoted as ϕ_D -- although ϕ_P and ϕ_D were 41.94 % and 42.05%, respectively. However, these values are presented as 41% and 43%, respectively, instead in the main paper.

Response: We thank the Reviewer for pointing out the mistake. The correction has now been made in the revised manuscript and Supplementary Information.

8) Equations inserted in the main paper should be numbered in order and referred to in the text.

Response: We thank the Reviewer for pointing out the omission. We have now added the equation numbers properly in the revised manuscript and Supplementary Information.

Reviewer #2 (Remarks to the Author):

The authors studied the effect of polarons-induced exciton quenching to understand the efficiency roll-off in TADF based OLEDs. To investigate SPA and TPA, drift-diffusion numerical modeling, exciton dissociation theory, kinetic exciton dynamics were introduced and successfully allowed understanding of bimolecular exciton kinetics. The manuscript was well organized, and the results of experiment and modeling were convincing.

Response: We are delighted and very grateful for the Reviewer's positive feedback and valuable comments.

However, some issues should be addressed before the publication.

1. In drift-diffusion modeling for determining the polaron density (actually field-depnt mobility in the modeling), was the trap-dynamics included? Active layer in the HOD included both mCP host and 4CzIPN dopant and hole trap was existed considering the HOMO level. Is there a reason to set the DOS as fitting parameter? I wonder the effect of DOS value to the result of the simulation.

Response: We thank the Reviewer's comment. Our primary objective for the drift-diffusion modelling was to plug-in the polaron densities under different applied voltages in order to obtain PL quenching as a function of polaron densities. Several factors can influence current density at different voltages. Since the whole work involved numerous parameters influencing OLED and HOD device properties, for the simplicity of the calculation the trap-dynamics were not included in the model. Introduction of trap-dynamics can slightly change the extracted values of annihilation constants. However, we believe the impact of this will be fairly small in the context of our work, and hence the final conclusions will not significantly change due to any minor variation in the annihilation rate constants.

In this work, the DOS (N_v) for organic semiconductor was assumed to be close to 10^{21} cm^{-3} [*J. Chem. Phys.* **119**, 2669-2679 (2003); *J. Appl. Phys.* **102**, 104503 (2007); *Phys. Rev. B* **81**, 035327 (2010)]. A small increase in current density was observed with the increase of DOS level.

Figure | Impact of DOS level. Experimental (circle) current density–voltage response from HOD and current density calculated from drift-diffusion model (solid line).

2. In EQE roll-off fitting (Fig.4), the charge balance factor (Y) was decreased from 1 to 0.1 with increasing the current density. I think this change was quite large. Considering the factor, recombination zone would be changed with driving voltage. How was the EL spectrum shift dependent on the voltage?

Response: The Reviewer has made a very important point here. In this study, only two transport layers were used to inject the holes and electrons into the emissive layer. As a consequence, the difference in energy barriers from the cathode and anode, and the unbalanced carrier mobility in the emissive layer can result in the decrease of charge balance factor in the devices [*J. Appl. Phys.* **89**, 4575-4586 (2001)]. Figure below shows the current density dependent spectra collected from OLEDs. There is a gradual shift in the EL spectra with the increase in current density, which may be due to the shift of recombination zone and the associated cavity effects. This supports the change in the charge balance factor and possibility of optical/outcoupling losses [*Org. Electron.* **70**, 219-226 (2019)] with increasing current densities.

Figure | Current density dependent EL. A blue shift in EL spectra with increasing current densities was observed. Inset shows the zoomed EL spectra, from which a shift of approximately 5 nm can be estimated.

3. The calculation results in Fig. 5 demonstrated that TPA was main contributor of the rolloff and TTA showed very low portion in triplet dynamics. I think it might be attributed that TTA coefficient was adopted from the literature and fixed and TPA was set as fitting parameter in simulation.

Response: We want to thank the Reviewer to point out this important issue. The TTA rate was adopted from literature as $5 \times 10^{-18} \text{ cm}^3 \text{ s}^{-1}$ [*Appl. Phys. Lett.* **113**, 083301 (2018)]. In this report, TTA was investigated in the same blend of mCP:5 wt% 4CzIPN as ours. Thus, for the convenience we have fixed the TTA rate in our study in order to calculate triplet dynamics. However, there are many recent works that have focused on the determination of triplet diffusion length and triplet diffusivity by applying different theoretical approximations and experimental techniques. Previously, triplet diffusivity of neat 4CzIPN was reported as $7.3 \times 10^{-9} \text{ cm}^2 \text{ s}^{-1}$ [*Adv. Mater.* **31**, 1804490 (2019)] and $6 \times 10^{-10} \text{ cm}^2 \text{ s}^{-1}$ [*Chem. Sci.* **12**, 1121-1125 (2021)] by adapting two separate measurement processes. Assuming TTA via diffusion-based Dexter transfer process, k_{TT} can be expressed as $k_{TT} = 4\pi R_a D_T$, where R_a is the annihilation radius and D_T is the triplet diffusivity. The value of R_a is generally assumed to be

1 nm [*Phys. Rev. B* **68**, 235212 (2003); *J. Phys. Chem. B* 107, 7696-7705 (2003); *Phys. Rev. B* **71**, 155204 (2005)] in organic semiconductors, though there have been reports of value as 4 nm [*Phys. Rev. B* **63**, 165213 (2001)] and greater [*Org. Electron.* **7**, 452-456 (2006)]. By selecting the higher value of R_a as 4 nm, we can estimate k_{TT} as $3.66 \times 10^{-14} \text{ cm}^3 \text{ s}^{-1}$ and $3.01 \times 10^{-15} \text{ cm}^3 \text{ s}^{-1}$ for the triplet diffusivity of $7.3 \times 10^{-9} \text{ cm}^2 \text{ s}^{-1}$ and $6 \times 10^{-10} \text{ cm}^2 \text{ s}^{-1}$, respectively. By adapting these values of k_{TT} and the values obtained for k_{TP} from OLED and HOD device studies in our work, we have calculated relative contributions for triplet deactivation pathways to investigate the impact of TTA and show in the figure below. Interestingly, with several orders of high k_{TT} values, the influence of TTA slightly increased from the calculations in the manuscript. Though the amount of contribution still remained insignificant in 4CzIPN OLEDs compared to other deactivation processes. The maximum contribution of TTA was obtained as 9% from $k_{TT} = 3.66 \times 10^{-14} \text{ cm}^3 \text{ s}^{-1}$ and $k_{TP} = 3.66 \times 10^{-14} \text{ cm}^3 \text{ s}^{-1}$.

Figure | Triplet dynamics. Relative contribution of TTA with different k_{TT} and k_{TP} values.

4. Were the simulations in the paper commercial program or in-house code? Brief information of the simulation should be included in the paper such as programming language and fitting algorithm for the readers.

Response: For this work, we have used simplified in-house code to numerically solve charge transport in HODs and the excitonic process in the TADF OLEDs. The detail description of the

model is given in the manuscript and Supplementary Information. Also, upon any reasonable request we will share more details or codes with the Readers.

5. I suggest that x-axis of fig.5 would be appropriate to be the current density or luminance for intuitive understanding since the light emission did not occurred in low voltage.

Response: We thank the Reviewer for the helpful suggestion. The modified singlet and triplet dynamics of OLEDs as a function current density have now been added to the manuscript (now Fig. 4 in the manuscript) with associated change in manuscript text.

Modified figure:

Fig. 4. Relative contributions of the different excitonic processes associated with TADF emission. Relative contributions are plotted for mCP:4CzIPN OLEDs as a function of current density **a** Deactivation of singlet excited state, **b** Deactivation of triplet excited state.

6. The molecular structure of BP4mPy was not included. Also, inclusion of device structure with frontier orbital energetics would be good to better understanding of the readers.

Response: We acknowledge the Reviewer's comment. The molecular structures of hole transport layer (PEDOT:PSS) and electron transport layer (BP4mPy) have now been added to the supplementary Fig 7b. The device structure of HOD and OLEDs with frontier orbital energetics being added to Supplementary Fig 2b, Fig 7a, Fig 10b and Fig 11a as addressed in our Response to Comment 4 of Reviewer 1.

The added molecular structures:

Supplementary Fig. 7 b | Molecular structures. Hole transport material poly(3,4-ethylenedioxythiophene):poly(styrenesulfonate) (PEDOT:PSS) and electron transport material 3,3',5,5'-tetra[*m*-pyridyl]-phen-3-yl]biphenyl (BP4mPy).

7. Some formats should be modified. Number of equations were wrong. The symbols of PLQY for prompt and delay in Si TABLE 1d were same. The structure of SI should be modified in terms of the sections or individual figures and tables. In terms of the figures and tables, all components should be addressed in the main manuscript.

Response: We thank the Reviewer for pointing out the mistake. We have now made the required correction in the revised manuscript and Supplementary Information. The structure of

the Supplementary Information has now also been modified according to the Reviewer's suggestion.

Reviewer #3 (Remarks to the Author):

Presented by Hasan et al., this paper managed to discuss polaron-induced exciton quenching in TADF-based organic light-emitting diodes (OLEDs). By utilizing steady-state photoluminescence (PL) and electroluminescence (EL) measurements, the authors attempted to quantify the quenching rates of singlet-polaron annihilation and triplet polaron annihilation then to analyze their influence on the OLEDs performance. As understanding quenching mechanisms in OLEDs plays crucial role in improving device efficiencies, this paper is of importance to the materials science community, thus qualified for being published in Nat. Commun. Nevertheless, several issues should be addressed properly before further consideration for being published. Comments and suggestions are listed below:

Response: We thank the Reviewer for careful evaluation of our work and positive comments. We have now made the relevant changes to the main text and supporting information based upon the concerns raised.

1. Apart from exciton-polaron annihilation, the quenching possibly arises from local heat or even from the decomposition of emitters under a high voltage. The authors should at least provide direct experimental evidence of polarons and comment on the influence of excitation (or current) density and lattice heating effect on their measurements.

Response: We agree with the Reviewer that apart from exciton-exciton and exciton-polaron annihilations heat induced quenching and decomposition of emitter reduce exciton density can impact under high voltage conditions. Previous studies with fluorescent OLEDs suggest that heat induced quenching rate (k_{th}) can be determined by considering the power dissipation (JV) in OLED operation [*Appl. Phys. Lett.* **86**, 213506 (2005); *J. Appl. Phys.* **126**, 185501 (2019)]. However, in TADF OLEDs, the implication of heat induced quenching is far more complex than fluorescent/phosphorescent OLEDs. Since several reports have shown that increase in temperature can significantly change RISC rate [*Nat. Mater.*, **18**, 1084-1090 (2019); *Phys. status solidi A*, **217**, 1900616 (2020)] that can alter the singlet and triplet dynamics and the overall impact of annihilation. This further complicates the theoretical model and resultantly

becomes less pragmatic in understanding the polaron-effect on EQE roll off. It suggests that our extracted polaron induced quenching rate may be slightly overestimated. However, any increase in RISC can cancel out the heat induced losses.

Many previous studies had experimentally drawn conclusions that PL intensity drop under electrical bias is predominantly due to polaron-induced exciton quenching and lacked mentioning about heat-induced annihilation [*Phys. Rev. B* **75**, 125328 (2007); *Adv. Funct. Mater.* **23**, 5420-5428 (2013); *Appl. Phys. Lett.* **105**, 143303 (2014); *Adv. Funct. Mater.* **24**, 6074-6080 (2014); *J. Phys. Chem. C* **119**, 7631-7636 (2015)]. We do agree with the possibility of the heat-induced quenching, though it is extremely challenging to separate polaron induced quenching from heat induced quenching in a TADF system. However, any decomposition of emitter molecule that can cause change in emission spectra was not observed in our investigation, and hence we found it appropriate to proceed in the analysis of EQE roll-off without taking thermal effects into the picture.

2. The authors should comment on why a dual excitation setup was specifically used for this study.

Response: Exciton-exciton interaction (SSA, STA, and TTA) and exciton-polaron interaction (SPA and TPA) are spin-allowed processes in TADF and thus can result in the reduction of efficiency in OLEDs. It is quite challenging to separate these processes under steady-state OLED operation, specifically in TADF as they can all simultaneously occur in high current densities. Without separating annihilation processes appropriately, any measurements of annihilation rates can lead to overestimation of the value. The benefit of using a dual excitation setup for the investigation of polaron-induced quenching is that it can be used to separate exciton-polaron quenching from exciton-exciton interactions. Since this setup can constantly produce same number of excitons *via* optical pumping, while any change in the PL intensity collected from the device can be assigned due to exciton-polaron interactions. In addition, under reverse bias in OLEDs, it can be utilised to block any charge injection in order to collect change in PL intensity due to different applied electric fields. In this way, more accurate estimation of exciton-polaron quenching rate and field-induced dissociation rate can be obtained by using dual excitation setup.

3. In Supplementary Fig.4b, the prompt and delay lifetimes are obtained through tail fit instead of deconvolution fit. Since the decay profile (purple dots) is already a convolution between excitation source and population decay, it is recommended to perform deconvolution fit with instrumental response function to obtain an accurate prompt decay time, Φ_P and Φ_D .

Response: We would like to thank the Reviewer for pointing out this important issue. TCSPC measurements to extract the lifetimes was performed with Jobin-Yvon Fluorolog-3, by exciting the samples at an excitation wavelength of 372 nm, generated by a pulsed nano-LED and an instrument response of about 1 ns. Figure below shows the instrument response function. In our work, $IRF \ll$ prompt and delayed lifetime of 4CzIPN. As a consequence, we believe that the extracted lifetime will not significantly differ from the deconvolution fit. Figure of IRF has now been added to Supplementary Fig. 5.

Supplementary Fig. 5 | Photophysical properties of mCP:4CzIPN blend. Excited state decay of the mCP:4CzIPN blend under optical excitation. Solid line represents the bi-exponential fit for the decay. inset shows the instrument response function (IRF).

4. The parameters, k_{SS} , k_{SP} , and k_{TP} , are rate constants, which should be investigated by time-resolved measurements (PL and/or EL) instead of merely by the fitting results from steady-state approaches.

Response: We do agree with the Reviewer. Both steady-state approach and time-resolved approaches are used extensively to estimate annihilation rate constants in organic semiconductor. Both approaches have its own benefit and drawback. Time-resolved approach can be used to separate quenching processes in timescale. However, time resolved approaches

often gets complicated by dispersion effects, which can lead to time-dependent rate constants [*Chem. Phys. Lett.* **652**, 142 (2016); *Adv. Opt. Mater.* 2100249 (2021)]. In addition, a previous report also suggested that steady-state approach simplifies rate equations in such a way that it is more suitable for the extraction of annihilation rate constants in TADF emitters [*Appl Phys. Lett.* **113**, 083301 (2018)]. Furthermore, all the measurements were performed under same steady-state conditions, that allowed us to co-relate all the results more effectively.

5. I recommend the authors to move Fig. 1 to the supplementary information, as it appears that the purpose of simulating current density-voltage response is only to obtain the hole mobility and characteristic field value, both of which are not further evaluated.

Response: We appreciate the Reviewer's suggestion. In the revised manuscript Fig. 1 has now been moved to the Supplementary Information as Supplementary Fig. 6a.

6. What is the base for the authors to attribute the field dependent PL reduction solely to the dissociation of singlet excitons without considering the dissociation of triplet excitons? The excited-state population of the singlet and triplet excitons should be roughly at the same level, since the prompt and delayed efficiency, which corresponds to the singlet and triplet excitons, respectively, share a similar contribution to the QY.

Response: We do agree with the Reviewer that both singlet and triplet excitons can dissociate under electric field and can contribute to the field dependent PL reduction. In our revised manuscript, we have added a new term $R(f)T$ to the triplet rate equation. To further investigate this, we have performed a separate experiment under transient optical excitation. The OLEDs under reverse bias were excited with a 3.5 ns pulsed laser and the PL response was collected via a photomultiplier tube. The newly added Supplementary Fig. 8 shows the time-resolved PL response collected at 0 and 15 V applied bias. It is apparent from Supplementary Fig. 8a that there is a drop only in the prompt intensity under high applied bias, indicating exciton dissociation under high electric field. Interestingly, the normalized plot in Supplementary Fig. 8b shows there is an increase of contribution in the delayed component under high electric fields. This can either happen either due to electric field modifying singlet-triplet energy splitting such a way that intrinsic RISC rate (k_{RISC}) is increasing or the dissociation of singlet excitons far exceeds the triplets. However, any changes in the lifetime of the delayed

component or in emission spectrum were not observed, suggesting latter process might be more probable.

To take this effect in the account, we have now added lines in the manuscript and figures in the Supplementary Information.

Added figure and text in the Supplementary Information and manuscript:

Supplementary Fig. 9 | Time-resolved PL response collected from 4CzIPN OLEDs. a PL response collected from OLEDs under reverse bias with pulsed optical excitation. The decrease in intensity under high voltage condition (i.e., 15 V) is assumed to be due to the field induced quenching. **b** Normalized PL intensity as function of time. There is an increase of contribution from the delayed component, assumed to be due to less amount of quenching from triplet states.

“To explore the effect of electric field induced dissociation in singlet and triplet excitons, time-resolved PL responses from the OLED are collected with dual excitation shown in Supplementary Fig. 8. Interestingly, PL decays collected under external electric fields showed higher contribution from the delayed component compared to those without electric fields. Thus, triplet quenching is assumed to be insignificant, and $R(f)T$ should not influence the dynamics of singlet and triplet densities studied in this work.”

7. In main text, P.8, Eq. (3), the recombination zone (d) is set as 15 nm. A reference is required for this assumption.

Response: We acknowledge the Reviewer's comment. References have now been added to the revised manuscript to back our assumption. The following references have been now added; *Adv. Funct. Mater.* **24**, 6074-6080 (2014); *Org. Electron.* **14**, 2721-2726 (2013).

8. In main text, P.8, Eq. (4), a singlet generation ratio of 0.25 is assigned for both exciton generation and triplet-triplet annihilation based on spin statistics. However, according to previous TTA work on anthracene (*J. Am. Chem. Soc.* 1981, 103, 7159-7164), it is easily shown that the possibility of excited singlet pair generated from triplet pairs will not exceed a factor of 1/9. Thus the authors should reevaluate the coefficient used for kTT. Furthermore, the kTT term should be a plus sign as it stands for singlet generation. Likewise, in Eq. (5), why is the coefficient of kTT being $-(1+\alpha)$ instead of -1? (*Phys. Rev. B* 2002, 66, 035321)

Response: We thank the Reviewer for pointing this out. In some literature singlet generation ratio due to TTA process can be found to be α with a value of 0.25 [*Phy. Rev. B* **84**, 115208 (2011); *J. Appl. Phys.* **101**, 023107 (2007)]. However, we do believe a more appropriate model was present in *Nat. Commun.* **10**, 1-10 (2019) to include TTA process in the rate equations, where coefficient for TTA was quite similar to *J. Am. Chem. Soc.* **103**, 7159-7164 (1981). By adapting the rate equation presented at *Nat. Commun.* **10**, 1-10 (2019), we have now modified our excitonic model as

$$\frac{\partial P}{\partial t} = \frac{J(t)}{qd} - \gamma P^2, \quad (3)$$

$$\frac{\partial S}{\partial t} = Y(J)\alpha\gamma P^2 - k_S S - k_{ISC} S + k_{RISC} T - k_{SP} SP - k_{SS} S^2 - k_{ST} ST + \frac{\alpha}{2} k_{TT} T^2 - R(f)S, \quad (4)$$

$$\frac{\partial T}{\partial t} = Y(J)(1 - \alpha)\gamma P^2 - k_{RISC} T - k_T T + k_{ISC} S - k_{TP} TP - \frac{1+\alpha}{2} k_{TT} T^2 - R(f)T, \quad (5)$$

All the calculations in the manuscript are now updated using this set of equations.

We also would like to point out that the term k_{TT}' mentioned in *Phys. Rev. B* **66**, 035321 (2002) is represented as k_{TT} in our work. In *Phys. Rev. B* **66**, 035321 (2002), they have taken k_{TT} as coefficient $\times k_{TT}'$. The updated coefficient used in this work is very close to the value of the coefficient used in that paper.

9. The fitted kSP and kTP from PL (6.1×10^{-12} cm³s⁻¹ and 5.8×10^{-13} cm³s⁻¹, respectively) and EL measurements (2×10^{-11} cm³s⁻¹ and 9.1×10^{-13} cm³s⁻¹, respectively) for mCP:4CzIPN (also for ACRXTN OLED) is not reasonable to me. As the lifetime of singlet excitons is known to be much faster than that of triplet excitons, even disregarding the

population gap between singlet and triplet excitons, the chance for polarons to annihilate singlet excitons should be much lower than to annihilate triplet excitons. How do the authors explain the results that $k_{SP} > k_{TP}$?

Response: The Reviewer has made a very important point here. Interestingly, the magnitude of k_{SP} and k_{TP} is independent of singlet and triplet lifetime, respectively. Though the amount of quenching caused by an individual annihilation process is greatly dependent on excited state lifetimes. For example, in case of TPA, the amount of quenching/loss due to TPA is proportional to $k_{TP}TP$, where lifetime of triplet influences the loss mechanism *via* triplet density (T). High triplet lifetime can give rise to triplet density (triplet accumulation) over time, which can increase the amount of quenching with a low k_{TP} . However, the value of k_{TP} and k_{SP} depends on triplet and singlet diffusivity. The SPA and TPA predominantly occur via long-range Förster and short-range Dexter transfer processes, respectively. Triplet-polaron quenching requires a collision between the species, and hence the rate is much lower than singlet-polaron quenching. In general, Förster-transfer based annihilation process such as SSA, STA and SPA tends to have high-rate constant values compared to Dexter-transfer based annihilation constants such as TPA and TTA.

10. For the fitting processes to acquire k_{SP} and k_{TP} , the authors state that the STA and TTA rate constants were taken from the reference paper (*Appl. Phys. Lett.* 113, 083301 (2018)). Yet, this reference paper reported a series of simulations with different sets of STA and TTA values. Which set of values did the authors adopted in this manuscript? Please specify all the fixed values in the fit process at least in the supplementary information.

Response: We thank the Reviewer for raising a very valid point here. In the manuscript we have now specified the values of STA and TTA, taken from *Appl. Phys. Lett.* 113, 083301 (2018).

Modified line:

The STA and TTA rate constants were taken from the literature as $1 \times 10^{-11} \text{ cm}^3 \text{ s}^{-1}$ and $5 \times 10^{-18} \text{ cm}^3 \text{ s}^{-1}$, respectively¹⁸ and used to fit the EQE curve.

Minor points:

1. All equations should be properly labeled from Eq.(1) to Eq.(7).

Response: We thank the Reviewer for pointing out the omission. The correction has now been added to the revised manuscript and Supplementary Information.

REVIEWERS' COMMENTS

Reviewer #1 (Remarks to the Author):

The authors have addressed all my 11 numbered comments rather meticulously, convincingly, and comprehensively. Thus, I enthusiastically recommend publication of this revised manuscript in Nature Communications as is.

Reviewer #2 (Remarks to the Author):

This paper is ready for publication.

Reviewer #3 (Remarks to the Author):

All questions have been properly answered by the authors, including the insignificant field induced quenching on triplet exciton, corrections on TTA coefficients and rationalization on the numerical value of k_{SP} and k_{TP} . The current format is well organized with self-consistent experimental results and discussions, which deserve its publication on Nat. Commun. Further revision is not needed after editing on a typo and statement:

1. Main text, page 10, "The k_{SP} and k_{TP} found from non-doped ACRXTN HOD----- [$1.3 \times 10^{-12} \text{ cm}^3 \text{ s}^{-1}$] -----".
2. Main text, bottom of page 15, the statement "However, as the delayed lifetime----[inversely proportional] -----

Reviewers' comments:

Reviewer #1 (Remarks to the Author):

The authors have addressed all my 11 numbered comments rather meticulously, convincingly, and comprehensively. Thus, I enthusiastically recommend publication of this revised manuscript in Nature Communications as is.

Response: We thank the reviewer for recommending this manuscript for publication at Nature Communications. We are delighted and grateful for the Reviewer's positive feedback.

Reviewer #2 (Remarks to the Author):

This paper is ready for publication.

Response: We appreciate the reviewer for recommending this manuscript for publication. We are grateful that the referee has spent time reviewing and is satisfied with our responses.

Reviewer #3 (Remarks to the Author):

All questions have been properly answered by the authors, including the insignificant field induced quenching on triplet exciton, corrections on TTA coefficients and rationalization on the numerical value of k_{SP} and k_{TP} . The current format is well organized with self consistent experimental results and discussions, which deserve its publication on Nat. Commun. Further revision is not needed after editing on a typo and statement:

1. Main text, page 10, "The k_{SP} and k_{TP} found from non-doped ACRXTN HOD----- [1.3 $\times 10^{-12}$ cm³ s⁻¹] -----".
2. Main text, bottom of page 15, the statement "However, as the delayed lifetime---- [inversely proportional] -----

Response: We are delighted and grateful for the Reviewer's positive feedback on our revised manuscript. We want to thank the Reviewer for the valuable comments. The typo and statement mentioned by the reviewer were edited accordingly.

Modified lines:

1. The k_{SP} and k_{TP} found from non-doped ACRXTN HOD were $1.3 \times 10^{-12} \text{ cm}^3 \text{ s}^{-1}$
2. However, as the delayed lifetime is inversely proportional to the delayed efficiency and rate of reverse intersystem crossing.....